# Kinetic Modeling of Time-Dependent Enzyme Inhibition by Pre-Steady-State Analysis of Progress Curves: The Case Study of the Anti-Alzheimer’s Drug Galantamine

**DOI:** 10.3390/ijms23095072

**Published:** 2022-05-03

**Authors:** Doriano Lamba, Alessandro Pesaresi

**Affiliations:** 1Istituto di Cristallografia, Consiglio Nazionale delle Ricerche, Area Science Park, I-34149 Trieste, Italy; doriano.lamba@ic.cnr.it; 2Consorzio Interuniversitario, Istituto Nazionale Biostrutture e Biosistemi, I-00136 Rome, Italy

**Keywords:** acetylcholinesterase, galantamine, Alzheimer’s disease, time-dependent inhibition, residence-time, global fitting of progress curves

## Abstract

The Michaelis–Menten model of enzyme kinetic assumes the free ligand approximation, the steady-state approximation and the rapid equilibrium approximation. Analytical methods to model slow-binding inhibitors by the analysis of initial velocities have been developed but, due to their inherent complexity, they are seldom employed. In order to circumvent the complications that arise from the violation of the rapid equilibrium assumption, inhibition is commonly evaluated by pre-incubating the enzyme and the inhibitors so that, even for slow inhibitors, the binding equilibrium is established before the reaction is started. Here, we show that for long drug-target residence time inhibitors, the conventional analysis of initial velocities by the linear regression of double-reciprocal plots fails to provide a correct description of the inhibition mechanism. As a case study, the inhibition of acetylcholinesterase by galantamine, a drug approved for the symptomatic treatment of Alzheimer’s disease, is reported. For over 50 years, analysis based on the conventional steady-state model has overlooked the time-dependent nature of galantamine inhibition, leading to an erroneous assessment of the drug potency and, hence, to discrepancies between biochemical data and the pharmacological evidence. Re-examination of acetylcholinesterase inhibition by pre-steady state analysis of the reaction progress curves showed that the potency of galantamine has indeed been underestimated by a factor of ~100.

## 1. Introduction

The study of enzyme inhibition is an essential aspect of biochemistry with implications in many fields of life sciences, including mechanistic enzymology, drug design, pharmacology and the study of metabolism regulation [1]. In particular, the correct assessment of the inhibition constant (*K_i_*) and of the type of inhibition caused by a drug (Figure 1) are important pieces of information required to translate biochemical data into pharmacodynamics models [2,3,4]. Half-maximal inhibitory concentration (IC_50_), for example, is notoriously a very poor predictor of the pharmacodynamics properties of a drug [5]. It does not account for the inhibition mechanism nor for the inhibitors’ residence time, defined as the reciprocal of the dissociation constant rate (1/*k_off_*), which recently has emerged as a critical feature of effective drugs [6,7]. The *K_i_* determination through a proper steady-state kinetic analysis, on the contrary, can provide a reliable depiction of the inhibition mechanism and of the kinetic parameters, although at the cost of a more time consuming and labor-intensive experimentation. Some circumstances, however, can turn the determination of an inhibitor’s *K_i_* into a rather problematic task. From this perspective, here is presented the case of acetylcholinesterase inhibition by galantamine.

Acetylcholinesterase (AChE, EC. 3.1.1.7) catalyzes the fast hydrolysis of the neurotransmitter acetylcholine, playing a crucial role in the regulation of the synaptic cholinergic signal transmission. The alteration of cholinergic systems is involved in many neurodegenerative disorders, including Alzheimer’s disease [8], hence a huge effort has been devoted over the decades to discovering and developing AChE inhibitors that can contrast cognitive impairment. Rivastigmine, donepezil and galantamine (GAL, Figure 2) are the three AChE inhibitors currently marketed as anti-Alzheimer drugs. GAL is an alkaloid originally isolated from the plant *Galanthus nivalis*, whose anti-cholinesterase activity was first discovered in the 1950s [9] and it was approved for the treatment of mild-to-moderate Alzheimer’s symptoms in 2001 [10]. It is also an allosteric modulator of nicotinic cholinergic receptor potentiating cholinergic neurotransmission [10]. In the incessant search for more effective anti-Alzheimer’s treatments, the development of new GAL derivatives continues to be an active field of investigation [11,12,13].

The evidence gathered along with more than 50 years of biochemical research suggested that GAL is a moderate AChE inhibitor, being 50 to 500 times less potent than donepezil [14], whose *K_i_* is in the low nM range [15,16]. Despite its pharmacological relevance, however, published data on the inhibition of AChE by GAL are surprisingly inconsistent. The 15 entries present in the BRENDA database [17] for example report IC_50_ values spanning almost 3 orders of magnitude, from 0.25 µM [18] to 100 µM [19]. While there are plenty of studies reporting in vitro or ex vivo IC_50_ determination, thorough steady-state kinetic analysis are rather scarce. To the best of the author’s knowledge, there are only two papers on the inhibition of the human brain AChE (HuAChE). The first, dating to 1976, was originally written in German and reported a competitive inhibition with *K_i_* of 52 nM [20]. The second, made using a recombinant form of HuAChE, was published in 2003 and reported a ten-fold higher *K_i_* of 0.52 µm [21]. Studies on the brain enzyme of mouse and rats confirmed the competitive nature of AChE inhibition by GAL and evaluated the *K_i_* to be of 0.86 µM and 0.16 µM, respectively [22,23]. In contrast, the inhibition of AChE from *Torpedo californica* (*Tc*AChE), an important enzyme because it was the first, and for a long time the only one to have been crystallized [24], was reported to be of the mixed-type with a *K_i_* of 0.2 µM [25]. Furthermore, a comparative in vivo study on rats, mice and rabbits suggested that GAL is an AChE inhibitor that is only 3–15 times weaker than donepezil [26], adding further confusion to the picture.

Because AChEs are highly conserved enzymes [27], their diversity alone can hardly account for the observed inconsistency. The likely explanation instead lays in the time-dependent nature of the inhibition exerted by GAL. As it is shown hereinafter, both the binding and the unbinding of GAL to the AChE active site are slow; a combination which seems to have tricked the unaware experimenters, preventing the correct assessment of the steady-state inhibition parameters and causing a gross overestimation of both *K_i_*s and IC_50_s.

The conventional steady-state analysis of enzyme kinetic in fact relies on two main assumptions, that substrate and inhibitor are in concentrations much larger than the enzyme, and that the establishment of thermodynamic equilibrium between all the reactants occurs rapidly on the steady-state time scale. For certain inhibitors, however, one or both of these two conditions might not apply and consequently the conventional steady-state treatment results inadequate. This occurs, for example, in the case of tight-binding inhibitors, which due to *K*_i_ values of 10 nM or smaller, are likely to be less than the concentration of the enzyme, and for slow-binding inhibitors. Slow enzyme-inhibitor (E-I) association rates can arise in different ways [28,29]. If a simple one-step interaction is assumed, slow association can be the consequence of a slow binding mechanism (small *k_on_*). Large *k_on_* can also result in a slow association in the case of strong inhibitors, because of their very low concentration in the assay. In some cases, the E-I interaction is better described as a two-step process, with the rapid formation of an initial collision complex followed by a slow isomerization leading to a tight EI complex [30].

The typical reaction progress curves of slow-binding inhibitors show an initial burst followed, once that the EI complex has formed, by the fall off of the turnover rate to the steady-state inhibited rate. This implies that the initial velocities are a large overestimation of the actual steady-state rates. The proper study of slow inhibition requires the determination of three parameters, i.e., the initial burst rate, the final inhibited steady-state turnover rate and a term that accounts for the rate of onset of inhibition [28,31,32]. The procedure, however, entails non-trivial data acquisition and data analysis problems. For instance, due to the depletion of the substrate occurring along with the reaction, the detection of the steady-state rates after tens of seconds or even minutes after the beginning of the reactions, especially for data at low substrate concentrations, can be challenging or impossible.

On the other hand, if the slow onset of inhibition is overlooked and kinetic data are forced into the simpler conventional model, slopes and intercepts of the resulting plots do not behave in the expected ways and the analysis is very likely to give misleading results [31,33,34].

To avoid the troubles arising with slow-binders it is a common practice to pre-incubate enzyme and inhibitor so that the equilibrium is established before the reaction is started [35]. This, however, exposes the experimenter to the risk that, for substrate-initiated reactions, the initial velocities largely depend on the rate of dissociation of the E-I complex [31]. Indeed, although the term “time dependent inhibition” generally is intended only as a synonym of slow-onset inhibition, for reactions started after an extensive E-I pre-incubation, slow dissociating inhibitors represent a further deviation from the steady-state assumptions whose study demands an equally complex analysis as in the case of slow inhibition [28,31] and whose implications surprisingly have not been thoroughly addressed.

A much more convenient and accurate way to study time-dependent inhibition is to drop the classic steady-state analysis based on the initial velocities in favor of the global fitting of the reaction progress curves [36]. This method, by being not based on any of the steady-state simplifying assumptions, allows us to draw information from the pre-steady-state phase of the reactions and to directly determine, beside the *K_i_*, the *on* and *off* microscopic rates for the EI complex formation [32].

The main focus of this paper is about the effects of slow inhibitor binding/unbinding on the conventional steady-state analysis of initial velocity data. Using a computational and theoretical approach we first show how, in the case of E-I pre-incubation, slow dissociation rates unavoidably cause competitive inhibition to seemingly look like mixed-type or noncompetitive. We than re-examine the inhibition of *Tc*AChE by GAL. The analysis of initial velocities showed the time-dependent nature of the inhibition providing a rationale that explains why the assessment of GAL potency has proven to be so problematic. Next, the global fitting of the reaction full time courses was performed with a two-fold intention. The first aim was to determine the actual GAL potency, which turned out to be at least 100-fold higher than what is generally believed. The second aim was to display the effectiveness and ease of use of this new method for the study of enzyme kinetics and inhibition. We provide a deep analysis of the molecular mechanisms leading to the slow AChE inhibition by GAL. Furthermore, we highlight the substantial contribution that the use of software for the direct simulation of enzyme kinetic can give to the advancement of biochemical and pharmacological research.

## 2. Results

### 2.1. Simulation of Enzyme Inhibition by Slow-Dissociating Inhibitors

To evaluate the impact of an inhibitor’s residence time on the assessment of the steady-state inhibition parameters, synthetic kinetic data were generated using the software KinTek Explorer [37]. Figure 3 shows the simulations of the activity of a hypothetical enzyme with a *K_m_* of 10 µM and a *k_cat_* of 1000 s^−1^ inhibited by three competitive inhibitors with an identical *K_i_* of 1 µM but with different dissociation rates. In order to simulate real experiments with the reactions started with the substrate, an extensive enzyme-inhibitor incubation time of 10^6^ s was allowed, and the reaction courses were lagged by 10 s to account for the dead time between the reaction start and the beginning of detection. In panels A–B are the curves and the relative data analysis for the fast unbinder (*k_off_* = 10^2^ s^−1^), in C–D those of an inhibitor with intermediate dissociation rate (*k_off_* = 3 × 10^−2^ s^−1^) and in E–F those of the slow unbinder (*k_off_* = 10^−4^ s^−1^). It is at first glance evident from the plot of initial velocities versus [S] (Figure 3A,C,E) how, despite the identical potency and mechanism of inhibition, inhibitors behave differently. Lineweaver–Burk plots confirm the differences, showing that for the fast unbinder the inhibition, as expected, is competitive (lines intersecting on the 1/V axis, Figure 3B), while the slow unbinder gives an inhibition that seems purely noncompetitive (line intersecting on the 1/[S] axis, Figure 3F). In general, for mixed-type inhibition, the lines of a double-reciprocal plot would be expected to intersect in a point at positive 1/V and at negative 1/[S] values. As shown in Figure 3D, however, the intermediate inhibitor data at low inhibitor concentrations are not consistent with those at higher concentrations. This results in an unexpected and characteristic plot, with intersections between lines that shift towards the 1/V axis as the [I] increases. However, if the competitive and uncompetitive contributions are uncoupled by analyzing data with the Dixon and Cornish–Bowden plot (see supporting information), for inhibitors with intermediate *k_off_*, the inhibition results are of the mixed type. Figure 3G shows how the apparent competitive (*K_ic_*) and uncompetitive (*K_iu_*) inhibition constants are affected by the dissociation constant rates. For fast dissociating inhibitors (*k_off_* > 3 × 10^−2^ s^−1^), the inhibition is competitive (*K_iu_ >>*
*K_ic_*). 

A gradual transition from competitive to noncompetitive occurs within *k_off_* values in the 3 × 10^−2^–3 × 10^−4^ s^−1^ range, where the inhibition appears to be mixed-type and *K_iu_* decreases with the inhibitor residence time. Then it turns purely noncompetitive, with *K_ic_* = *K_iu_*, for *k_off_* smaller than 3 × 10^−4^ s^−1^. The explanation of this apparent paradox is indeed trivial. When to start the reaction the substrate is added to the pre-incubated enzyme-inhibitor mixture, its binding causes a shift in the EI equilibrium according to the mass–action principle. For example, if the enzyme is incubated with an inhibitor concentration equal to *K_i_*, by definition at the equilibrium half of the total enzyme (e_0_) will be in the EI form, hence [EI] = [E] = 0.5 e_0_. If the reaction is started by adding the substrate in concentration equal to *K_m_*, a new equilibrium will be attained such that [EI] = [E] = [ES] = 0.33 e_0_. Hence, upon the reaction start, a partial dissociation of the EI complex takes place, which causes a corresponding apparent enzyme activation. In the presence of a competitive inhibitor the steady-state initial rate (v_ss_) is given by:$$v_{ss}=V_{max}\frac{\left[S\right]}{\left[S\right]+K_{m}\left(1+\frac{\left[I\right]}{K_{i}}\right)}.$$

If the enzyme is pre-incubated with the inhibitor, at time t_0_ when the *S* is added to start the reaction, the portion of E available to interact with *S* is a fraction of the total enzyme equal to:$$\frac{1}{1+\frac{\left[I\right]}{K_{i}}};$$

Hence, if the EI dissociation is slow, the actual initial limiting velocity (*V_imax_*) is:$$V_{imax}=V_{max}\left(\frac{1}{1+\frac{1}{K_{i}}}\right),$$
and by assuming an instantaneous E-S equilibrium, the measured initial velocity (*V_meas_*) is:$$v_{meas}=V_{max}\left(\frac{1}{1+\frac{\left[I\right]}{K_{i}}}\right)\frac{\left[S\right]}{\left[S\right]+K_{m}},$$
which corresponds to the equation for the pure non-competitive inhibition (*K_iu_* = *K_ic_*). For moderately slow EI dissociation rates, the measured initial velocities will be intermediate between *v_ss_* and *v_meas_*, leading to an apparent mixed-type inhibition (*K_ic_* < *K_iu_* < ~20 *K_ic_*).

Because the monitoring of the reaction has a dead time that generally is not less than ~10 s (unless a stopped flow apparatus is used), for fast dissociating inhibitors (i.e., *k_off_* larger than 1 s^−1^), the dissociation is not detected and it does not affect the measure of the initial velocities. On the contrary, if the dissociation occurs on a time scale similar to or larger than the typical dead time of the technique employed to monitor the reaction progress, it will be detected as an apparent enzyme activation. In this case, the recorded initial velocities underestimate the steady-state rates and the progress curves display an upward curvature resulting from the combination of EI dissociation and of substrate depletion.

This mechanism can be best visualized by the time course of the [ES] as a function or substrate concentrations (Figure 3H). Because the reaction rate is given by [ES] *k_cat_*, changes over time of [ES] are directly reflected in the slope of the progress curves. Moreover, because the mass–action effect is larger at high S and/or I concentration, data at low [I] and [S] are not consistent with those at a higher concentration. Hence, for *k_off_* resulting in the mixed inhibition range, the plots are difficult to interpret, the inhibition mechanism is very likely to be mistaken and, in general, the outcome of data analysis is dependent on how the experiment is carried out (reagents concentration used, dead time, time of curve integration). The time required for the recovery of the full enzyme activity is dependent only on *k_off_* and it can easily be calculated that the dissociation of ~95% of the bound inhibitor requires 3.5 times the residence time. Hence, once the reaction is started, a hypothetical inhibitor with *k_off_* of 10^−2^ s^−1^ would cause a slow reactivation that is completed in ~6 min (Figure 3H). For a proper characterization of the inhibition mechanism, steady-state turnover rates should be recorded after 6 min from the reaction start and after ~1 h for an inhibitor with *k_off_* of 10^−3^ s^−1^. However, even in that case, the detection of the actual steady-state initial velocities would remain impossible due to the concomitant substrate depletion, making the proper steady-state treatment of these kinetic data highly problematic.

### 2.2. Steady-State Analysis of AChE Inhibition by GAL

Figure 4 reports the progress curves for the hydrolysis of acetylthiocholine (ATCh) catalyzed by the *Tc*AChE in the absence or presence of GAL. If the reactions are started adding the enzyme as the last reagent, initial velocities are substantially independent from the inhibitor concentration (Figure 4A), which is diagnostic for slow-inhibition. As expected for a second order reaction, the rate of the EI formation is proportional to the GAL concentration and the onset of the inhibited steady-state rates is delayed accordingly. Conversely, if the reactions are started with the substrate after an E-I pre-incubation of 20 min, the progress curves show an upward curvature which is diagnostic for the slow inhibitor dissociation (Figure 4B). In this case the initial velocities are an underestimation of the actual steady-state rates. 

Despite the non-compliance with the steady-state requirements these initial velocities are analyzed according to the Michaelis–Menten model. Depending on how the reactions are initiated, the result is either a large overestimation of the *K_i_* or the emergence of an apparent uncompetitive inhibition component. For enzyme-initiated reactions, the Lineweaver–Burk plot is consistent with competitive inhibition and with a *K_i_* of 122.8 ± 20.7 nM (Figure 4C), while for the reactions initiated by adding the substrate as the last reagent, the inhibition seems to be of the mixed-type with *K_ic_* of 18.0 ± 1.5 nM and *K_iu_* of 52.7 ± 2.7 nM (Figure 4D). It is worth noting that, while for the substrate-initiated reactions the initial velocities fit the mixed-type inhibition model, for the reaction started with the enzyme the fit is rather inaccurate. Lines do not intersect on the 1/V axis, as expected for competitive inhibition, but at positive 1/[S] value (Figure 4C). The reason is that the actual initial velocities, i.e., the velocities extrapolated at time t_0_, would fit the non-competitive model if the reactions are started with substrate (see previous chapter), but in the case of the enzyme-initiated reactions their analysis would fail to detect any inhibition. Hence, in the latter case, the extent of the observed inhibition is largely dependent on the way the experiment is being carried out. Particularly relevant are the reagent concentrations and the portion of the progress curves that are integrated to detect the initial velocities. The faster the detection, the weaker the resulting apparent inhibition will be.

### 2.3. Pre-Steady-State Analysis of AChE Inhibition by Fitting of the Full Progress Curves

The study of enzyme kinetics by simulations generated by numerical integration is a not-so-new method [38] that in the last ten years has been made accessible to the general biochemist thanks to the development of a new generation of software [37,39]. The analysis of the progress curves of *Tc*AChE inhibited by GAL was performed using the ENZO web application, implemented at www.enzo.cmm.ki.si [39], (accessed on 1 December 2021). This program is designed to generate differential equations from drawn reaction schemes and to subsequently fit the coefficients of these equations by a least-squares method, to reproduce the experimental data using a numerical integration algorithm. There is not yet a univocal definition of this method, and terms such as “global fitting of progress curves”, “numerical integration of progress curves” or “kinetic analysis by direct simulation” should be considered as synonyms, as each one depicts a different aspect of the analytical process. The study of inhibition is a two-step process. The catalytic properties of the enzyme have to be resolved before the study of the inhibited curves can be performed. In principle, if there is no product inhibition, direct simulation by data-fitting of full reaction time courses allows the determination of the Michaelian parameters from a single progress curve [36,40]. In the case of the ATCh hydrolysis by AChE, however, the final product evolved in the presence of the Ellman’s reagent, i.e., thiocholine-thionitrobenzoic acid (thiocholin-TNB), is a relatively potent competitive inhibitor of the enzyme [41]. Hence, to simultaneously determine the kinetic mechanism and product inhibition, the van Slyke–Cullen single-intermediate reaction scheme [42] (Figure 5, Appendix A) for substrate hydrolysis by *Tc*AChE was combined with a competitive inhibition step to account for product inhibition. Then, a further competitive inhibition mechanism step was added to the scheme to probe the inhibition by GAL (Figure 5, Appendix A).

To determine *k*_cat_/*K_m_*, *k*_cat_ (corresponding to *k_1_* and *k_2_* of the van Slyke–Cullen reaction scheme, respectively) and to assay the product inhibition, five progress curves measured in the presence of initial thiocholine-TNB concentration from 0 to ~150 µM were analyzed with ENZO (Figure 6A and Appendix A). *k*_cat_ and *k*_cat_/*K_m_* were evaluated to be 2965 ± 3 s^−1^ and 3.73 ± 0.01 × 10^8^ M^−1^s^−1^ respectively. For the reactions at higher initial thiocholine-TNB concentrations, the plateau is reached at longer times which indicates inhibition by the reaction product. The curves fit with a competitive inhibition mechanism and a *K_i_* for thiocholine-TNB of 21.54 ± 0.38 µM. These values were constrained in the subsequent analysis of *Tc*AChE inhibition by GAL. In the evaluation, the second-order binding rate constant (*k_3_*) was set to the diffusion rate-limited value of 2 × 10^8^ M^−1^s^−1^, so that only the constant rates for the binding/unbinding of GAL to the free enzyme (*k_5_*/*k_6_*) had to be evaluated, along with the initial substrate and AChE concentrations. Figure 6B and Appendix A show the fitting of the progress curves measured in the presence of GAL.

Reactions were started by adding the enzyme as the last reagent, i.e., without E-I pre-incubation. Analogously to what was found for AChEs from other sources, and in contrast to what has previously been reported for the enzyme form *Torpedo californica* [25], the curves of the GAL-inhibited *Tc*AChE are consistent with a pure competitive inhibition mechanism. The *K_i_*, however, was evaluated to be of 8.45 ± 0.23 nM, showing that for this enzyme the potency of GAL is 25-fold larger than what previous *K_i_* determinations suggested [25] and 230-fold larger than what can be deduced by the reported IC_50_ of 1.82 µM [43].

It is important to note here that the pre-steady-state phase of the inhibited progress curves contains information on the kinetic of binding of GAL to *Tc*AChE. By fitting the curves with ENZO this information is automatically used to explicitly determine the *on* and *off* microscopic rates. On the basis of reagents concentrations and of the time for the onset of the steady-state (see Appendix A), the *k_on_* was found to be of 1.9 × 10^6^ M^−1^s^−1^ and, consequently, the *k_off_* was calculated to be of 1.57 × 10^−2^ s^−1^, corresponding to a residence time of 63.8 s.

## 3. Discussion

The dissociation constant rate, *k_off_*, is a direct measure of the thermodynamic stability of the EI complex, while the second order rate *k_on_* reflects the energy activation barrier for the binding reaction and hence the microscopic probability of a productive encounter between E and I [44]. *k_on_* has an upper limit in the diffusion rate, which at room temperature is ~10^8^–10^9^ M^−1^s^−1^ [6]. So, the binding of GAL to *Tc*AChE herein reported, with a *k_on_* of 1.9 × 10^6^ M^−1^s^−1^ is roughly 2–3 orders of magnitude slower than what is expected for a diffusion limited interaction. As already suggested for other slow AChE inhibitors [29], the likely explanation resides in the relation between the size of the GAL molecule and the geometry of the AChE active site. In AChEs, the active site is positioned at the bottom of a narrow and deep gorge [24] (Figure 7A). A feature common to all AChEs is the presence, roughly half way down the gorge, of four aromatic residues (Tyr121, Phe290, Phe331, and Tyr 334 in TcAChE), generally referred to as the bottleneck, which narrow the passage of in- and out-bound molecules down to a solvent accessible section that in TcAChE is of 28 Å^2^ (calculated by CAVER [45], Figure 7B [46,47,48,49]). GAL is a rigid and rather bulky molecule with dimensions of 8.9 × 6.5 × 3.0 Å. Hence, in order for GAL to reach its binding site *Tc*AChE has to undergo a significant conformational rearrangement. Because the motions of amino acid side chains and/or of secondary structure elements within proteins occur at a much slower rate than free diffusion [50], the inhibitor passage through the bottleneck likely constitutes the rate limiting step of the GAL-*Tc*AChE interaction. Recently, employing a surface plasmon resonance (SPR)-based assay, Fabini et al. have determined the constant rates for the binding to the human AChE of several inhibitors [51]. The *k_on_* and *k_off_* for GAL were evaluated to be 9.5 × 10^4^ M^−1^s^−1^, and 2.5 × 10^−2^ s^−1^ respectively, corresponding to a residence time of 40.3 s and to a *K_i_* of 0.26 µM. While the off constant rate is consistent with the *k_off_* that we have found for the enzyme of *Torpedo californica*, the *k_on_* is 20 times slower. A discrepancy that, however, seems to collide with the structural evidence available. Indeed, in HuAChE, the bottleneck is considerably narrower than in *Tc*AChE (its solvent accessible section is of 14 Å^2^, Figure 7C), suggesting that the extent of the structural distortion it has to undergo to allow the GAL through the gorge has to be larger, and hence that the binding should be equally as slow as or possibly slower than for *Tc*AChE. The likely explanation, supported by a rather rich literature, is that the chemical modifications and the consequent conformational constraints occurring when a protein is immobilized on the SPR sensor chip interfere with its dynamic properties, primarily affecting the detection of the *k_on_* [52,53]. The main conclusion that can be drawn from SPR data is the confirmation of the time-dependent nature of AChEs inhibition by GAL.

Because AChEs are highly conserved enzymes, the identification of a structural–activity relationship linking the crystallographic structures of human and *Torpedo californica* enzymes to the inhibition mechanism strongly suggests that the slow binding and unbinding of GAL are a general feature pertaining to all AChE isozymes. This, in turn, is the likely cause of both the large overestimation and the large variance observed within the published IC_50_s and *K_i_*s. In contrast to the accepted belief, mostly based on unreliable IC_50_ measurements, that GAL is a much weaker inhibitor of AChE than donepezil, here we proved that these two anti-Alzheimer’s drugs indeed share a very similar potency. This is also consistent with GAL and donepezil being used at similar therapeutic doses (16–24 mg/day and 5–10 mg/day, respectively) [54].

The data presented in this work support two conclusions. The first, of general relevance, is that a preincubating enzyme and inhibitor is not a safe way to study enzyme inhibition as this might cause, in the case of long residence time, to mistake a competitive inhibitor for a mixed-type or noncompetititve inhibitor. The second, which is specifically relevant to the field of anti-Alzheimer’s drug development, is that GAL is at least 100 times more potent than what has for many years been reported and hence that its potency is comparable to that of donepezil.

Furthermore, there is another general consideration that can be drawn and that we would like to share with the biochemist and medicinal chemist communities.

It is becoming increasingly clear that oversights similar to those regarding GAL and AChE are a rather common matter in biochemistry. The concern is such that efforts are being made to counteract the problem, for example, by developing software that can help to standardize and validate the analytical procedures [55] and databases that foster the accurate reporting of enzyme kinetic data [56]. However, as long as mechanisms that deviate from the steady-state model continue to be analyzed using methods that assume and require a steady state, any standardization attempt will only result in the standardization of errors.

A common source of data inaccuracy is the widespread use of dose–response experiments. IC_50_s are known to deliver a limited amount of information and it has been stressed many times that they should be handled with much caution [5,57,58]. Nonetheless, because dose–response analysis enables the assessment of ligand potency with a single (or very few) measurements, IC_50_s are regarded as an indispensable tool in drug design, where large compound libraries have to be screened [59]. Understandably, the labor-intensive analytical methods envisaged by enzymologists to study certain complex mechanisms can hardly be implemented in high-throughput screening methodologies and do not fit well with the needs of pharmacological research. At the same time, the pharmacodynamic models used to translate biochemical into pharmacological data and to predict the properties of potential drugs, are destined to fail if the biochemical inputs are too inaccurate [44,60,61]. It is important to remark that the overuse of IC_50_ does not pertain exclusively to medicinal chemistry. The tendency to oversimplify kinetic analysis seems to broadly affect all those fields of research where the study of enzyme activity is still central. See, as an example, the questionable habit to report *K_i_* derived by IC_50_ through the Cheng–Prusoff equation rather than by an actual steady-state analysis. The evidence herein reported shows once again how misleading the output of kinetic studies can be if oversimplified analytical approaches are employed. Methods that properly account for the deviations from the Michaelis–Menten model have indeed been devised but due to the complexity of the data analysis involved they have generally failed to turn into common real-life laboratory practices and, as a matter of fact, the accuracy that more thorough analysis might deliver is very often traded off with the ease of dose–response analysis.

In this context, the substantial benefit brought about by the development of computer programs for the assessment of enzyme kinetic by the global fitting of full reaction time-courses should be mentioned [62]. Notably, once the mechanism of the target enzyme has been resolved, the analysis of a single progress curve measured in the presence of the inhibitor is sufficient to provide an accurate *K_i_* determination and to discriminate between the possible inhibition mechanisms [63] (for a two-substrate enzyme, two curves at varying substrates concentrations would be required), hence abolishing the convenience of the IC_50_ measure over thorough analysis and *K_i_* determination. A deeper discussion of the principle and of the technicalities involved in this analytical method are beyond the scope of this work, and we refer the readers to the cited specialized literature. In general, the direct simulation of enzymatic reactions streamlines the otherwise difficult study of mechanisms that deviate from the steady-state assumptions, as are the cases, quite common in pharmacology, of tight-binders and time-dependent inhibitors. By facilitating a deeper and more thorough understanding of the interaction between pharmacological targets and lead compounds this new approach to enzyme kinetics might contribute to alleviating the high failure rate of drug candidates in clinical trials [64,65]. However, unfortunately, although software such as KinTek Explorer and ENZO has been released more than 10 years ago, they are still not very popular among biochemists, in part because their use implies the acquisition of new non-trivial skills. 

For the sake of the quality and reliability of both biochemical and pharmacological research, the authors call for a more resolute joint effort should be undertaken by teachers, scientists and the editors of scientific journals to promote the switch from the 100-years old initial velocities method to the computer-assisted global fitting of progress curves. This would concomitantly discourage the widespread use of the often-misleading IC_50_s.

## 4. Material and Methods

### 4.1. Generation and Analysis of Synthetic Kinetic Data

Synthetic data were generated with the software KinTek Explorer [37]. The program, developed to solve enzyme kinetics by the global fitting of the reaction full time-courses through numerical integration, implements a module which allows us to compute synthetic data once that a mechanism and initial conditions are given. To simulate the inhibition of an enzyme with *K_m_* of 10 µM by a competitive inhibitor with *K_i_* of 1 µM, *k_1_* and *k_-1_*, the constant rates for the association and dissociation of the ES complex were set to 10^8^ M^−1^s^−1^ and 1 s^−1^, respectively, and the catalytic constant *K_cat_* was set to 1000 s^−1^. The ratio between the *k_off_* and *k_on_* constant rates for the EI formation was set to 10^−6^ M. The *k_off_* value was varied in the 10^−5^–10^3^ s^−1^ range to simulate slow or fast dissociating inhibitors. Enzyme concentration was set to 100 pM, the concentrations of substrate were 1, 2, 5, 10, 20, 50, 100, 150, 200 µM and those of the inhibitor were 0.5, 1, 2, 5, 10, 20 and 50 µM. An extensive E-I pre-incubation of 10^6^ s was allowed to ensure that the thermodynamic equilibrium was attained before the start of the reaction. To simulate the dead time between the mixing of reagents and the recording of reactions, initial velocities were detected after 10 s from the start.

Double reciprocal plots were assessed by a weighted least-squares analysis that assumed the variance of the velocity (*v*) to be a constant percentage of *v* for the entire dataset. The calculation of the competitive inhibitor constant (*K_ic_*) value was carried out by re-plotting slopes of lines from the Lineweaver–Burk plot versus the inhibitor concentration and *K_ic_* was determined as the intersect on the negative x-axis (Dixon plot, Appendix A). The apparent uncompetitive *K_iu_* (dissociation constant for the enzyme–substrate–inhibitor complex) value was determined by plotting the apparent 1/*V_max_* versus inhibitor concentration (Cornish–Bowden plot, Appendix A). Data analyses were performed with LibreOffice Calc 7.2.

### 4.2. Chemicals

*Tc*AChE was isolated and purified as previously described [66], except for the affinity chromatography ligand, mono-(aminocaproyl)-p-aminophenyltrimethylammonium. Galantamine (purity > 99%) was from Sopharma AD (Sofia, Bulgaria). Dithio-bis-di-nitro benzoic acid (DTNB), acetylthiocholine iodide and all other chemicals were from Sigma-Aldrich (Milano, Italy). 

### 4.3. Measure of Enzymatic Activity

The enzymatic activity of *Tc*AChE was evaluated spectrophotometrically by Ellman’s method [67] using a Ultrospec 7000 (Milano, Italy) double beam spectrophotometer. Measures were carried out at room temperature in 25 mM K-phosphate buffer at pH 7.0 and 340 µM DTNB. The formation of the reaction product thiocholine-TNB was followed, measuring the absorbance change at 412 nm (ε 13,600 M^−1^cm^−1^).

### 4.4. Steady-State Inhibition Analysis

Reactions catalyzed by *Tc*AChE were measured in the absence and presence of galantamine for 60 s. For substrate-initiated reactions, 50 pM of enzyme was pre-incubated with GAL for 20 min. For the enzyme-initiated reactions, the enzyme concentration was 200 pM. Reciprocal plots of 1/velocity versus 1/[substrate] were constructed at substrate concentration in the 10–200 µM range. Data points are average values of three replicates. Four concentrations of inhibitors were selected for each assay: 50, 100, 200 and 400 nM for the enzyme-initiated reactions, and 5, 10, 20, 30 nM for the reaction started with the substrate. *K_ic_* and *K_iu_* were determined by Dixon plot and Cornish–Bowden plot, respectively (see Appendix A).

### 4.5. Kinetic Analysis of Reaction Progress Curves

Two distinct sets of progress curves were measured in triplicate either in the absence or in the presence of GAL. The analysis was performed using the ENZO web application. To evaluate the Michaelian parameters of the *Tc*AChE, the hydrolysis of 35 µM acetylthiocholine by ~200 pM enzyme was monitored for 15 min. To assay the product inhibition by thiocholine-TNB, five consecutive reaction time courses were measured in the same 1 mL cuvette by adding 2 µL of a 20 mM fresh substrate solution once the previous reaction had come to completion so that four concentrations from 35 to 150 µM of the inhibitory thiocholine-TNB compound were assayed.

For the determination of GAL inhibition, the reactions with 200 pM enzyme and 50 µM acetylthiocholine were monitored for up to 33 min. Galantamine concentrations were 15, 30, 60, 120 and 180 nM.

## Figures and Tables

**Figure 1 ijms-23-05072-f001:**
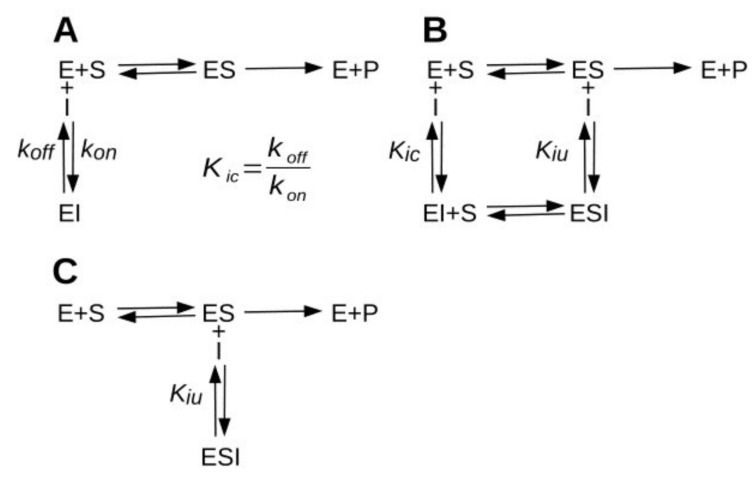
Inhibition mechanisms. (**A**) Competitive inhibition. The inhibition constant (*K_ic_*) is the dissociation constant for the formation of the EI complex. *K_ic_* corresponds to the ratio between the *off* and *on* constant rates. (**B**) Mixed-type inhibition. The inhibitor binds to both the free enzyme and to the enzyme-substrate complex (ES). The uncompetitive inhibition constant (*K_iu_*) is the dissociation constant for the formation of the trimolecular complex ES-I. If *K_ic_* = *K_iu_* the inhibition is defined as noncompetitive. (**C**) Uncompetitive inhibition mechanism.

**Figure 2 ijms-23-05072-f002:**
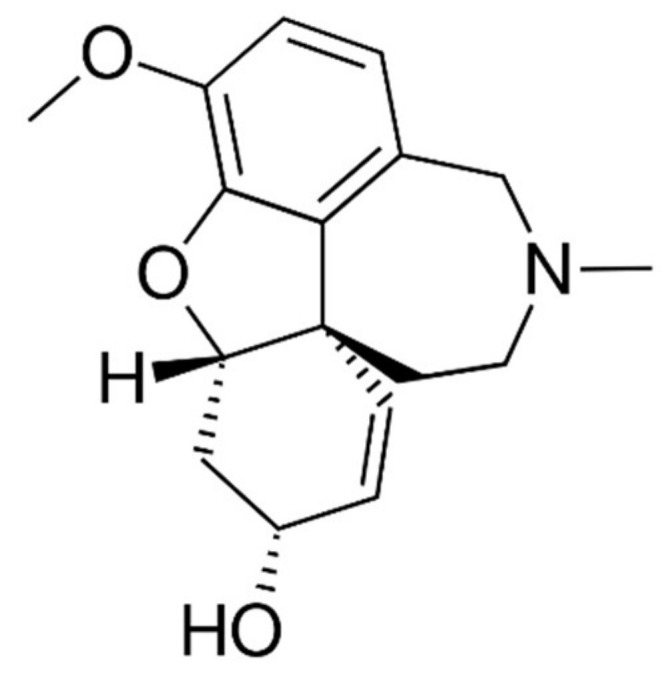
Molecular structure of galantamine.

**Figure 3 ijms-23-05072-f003:**
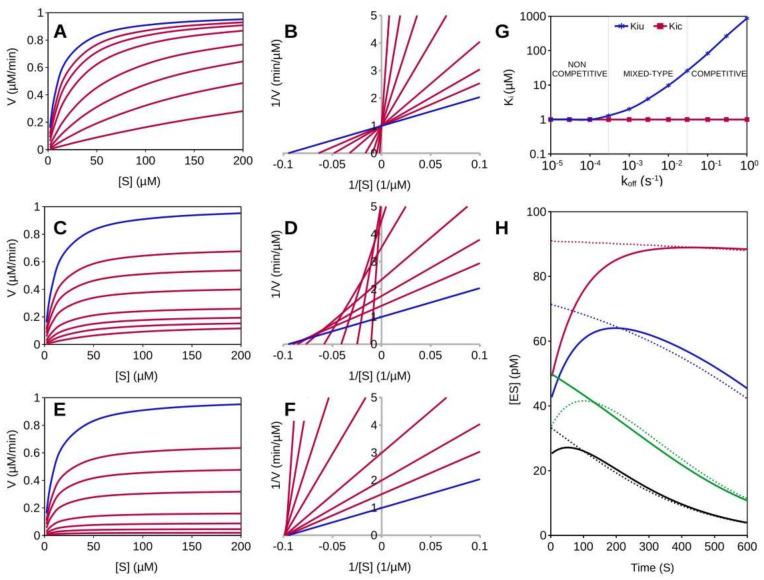
Analysis of synthetic kinetic data. Initial velocities were computed for an enzyme with *K_m_* = 10 µM and *k_cat_* = 1000 s^−1^ inhibited by three different competitive inhibitors with *K_i_* = 1 µM. An extensive E-I pre-incubation of 10^6^ s was allowed before the start of the reaction with substrate. (**A**,**C**,**E**) Initial velocity vs. [S] curves; for all curves [E] = 100 pM and [S] = 2, 5, 10, 20, 50, 100, 150, 200 µM. For each inhibitor [I] = 0, 0.5, 1, 2, 5, 10, 20 and 50 µM. Blue traces are for non-inhibited reactions. (**A**) fast unbinder (*k_off_* = 10^2^ s^−1^), (**C**) intermediate unbinder (*k_off_* = 3 × 10^−2^ s^−1^), (**E**) slow unbinder (*k_off_* = 10^−4^ s^−1^). (**B**,**D**,**F**) Lineweaver–Burk plots; for the fast dissociating inhibitor the inhibition is pure competitive (**B**) while the slow unbinder gives a plot typical of noncompetitive inhibition (**F**). For the intermediate inhibitor (**D**) a characteristic plot is obtained with an intersection between lines that shifts toward the 1/V axis as the [I] increases. For data point at low [I], the shift toward 1/V axis is less pronounced and the plot is consistent with mixed-type inhibition. (**G**) Emergence of the apparent uncompetitive inhibition component as a function of *k_off_*; for *k_off_* larger than 3 × 10^−2^ s^−1^ the inhibition as expected is competitive (*K_iu_* >> *K_ic_*). For *k_off_* smaller than 3 × 10^−3^ s^−1^ the inhibition seems pure noncompetitive (*K_iu_* = *K_ic_*). Artifacts suggesting a mixed-type inhibition occur for *k_off_* in the range ~3 × 10^−4^–3 × 10^−2^ s^−1^, where the corresponding apparent *K_iu_* value varies from 1 *K_ic_* to ~20 *K_ic_*. (**H**) Time course of the ES complex formation for substrate initiated reactions. The enzyme was pre-incubated with a slow dissociating inhibitor (*k_off_* = 10^−2^ s^−1^, continue lines) or with a fast dissociating inhibitor (*k_off_* = 10^3^ s^−1^, dashed lines). For all curves [E] = 100 pM and [I] = 1 µM. Four different substrate concentrations were assayed, 10 µM (black lines), 20 µM (green lines), 50 µM (blue lines) and 200 µM (red lines). It should be noted how the discrepancy between continuous and dashed lines at time t_0_ are larger for the curves at high [S]. For reactions at saturating initial [S] (red continue line), the velocity approaches the steady-state rate (red dashed line) after ~350 s of reaction. However, for reactions at lower initial [S] due to substrate depletion, the steady-state rate cannot be detected.

**Figure 4 ijms-23-05072-f004:**
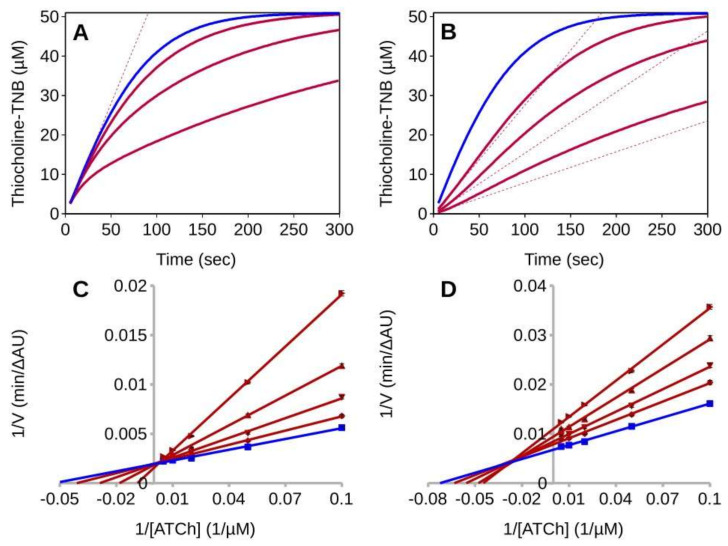
Steady−state analysis of TcAChE inhibition by GAL. (**A**,**B**) Time course of product formation for *Tc*AChE (200 pM) and ATCh (50 µM) in the absence (blue line) or presence (red lines) of GAL (30, 60, 180 nM). Dashed lines represent the slope of the first 10 s of reaction. (**A**) Reactions started with the enzyme (i.e., with no E-I pre-incubation). The initial rates are substantially unaffected by the presence of GAL and are a large overestimation the steady-state inhibited rates, which are only attained after 30–60 s from the start of the reactions. (**B**) Reactions started with substrate after a 20 min of E-I pre-incubation. Curves show an upward curvature so that initial velocities underestimate the steady-state turn-over rates. (**C**,**D**) Overlaid Lineweaver-Burk reciprocal plots of the TcAChE initial velocity (V) at increasing substrate concentrations ([ATCh] = 0–200 µM) in the absence and in the presence of inhibitors. Data points are average values of three replicates. Lines were derived from a weighted least-squares analysis of the data points. (**C**) Reaction started with enzyme (200 pM), in the absence or presence of GAL (0, 50, 100, 200, 400 nM). (**D**) Reaction started with substrate after an E-I pre-incubation time of 20 min in the absence (blue lines) or presence (red lines) of GAL (5, 10, 20, 30 nM). Enzyme concentration was 50 pM.

**Figure 5 ijms-23-05072-f005:**
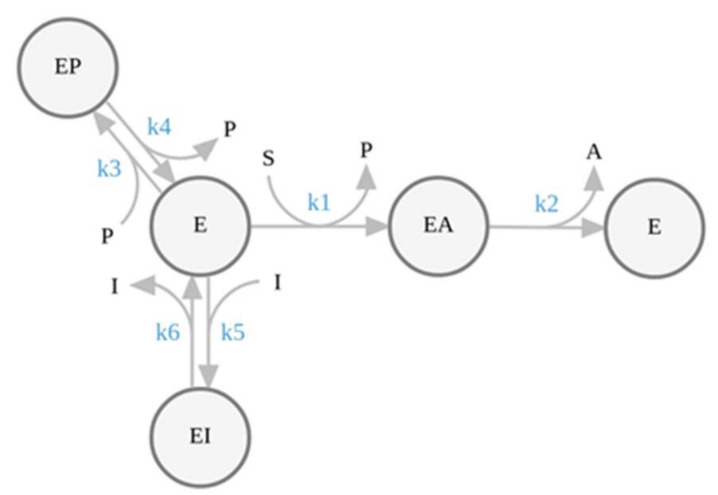
Mechanims of AChE inhibition by GAL. The hydrolysis of ACh (S) by AChE is a two-step reaction. The first step lead to the release of choline (P) and formation of the acylated enzyme (EA). In the second step the acyl-enzyme is resolved with release of acetate (A) and formation of free enzyme (E). P rapidly reacts stechiometrically with the Ellman’s reagent to form thiocholine-TNB (the detected specie), which competitively inhibits AChE. I is the inhibitor (galantamine). k_1_ is an apparent bimolecular constant rate equivalent to *k_cat_*/*K_m_* and k_2_ is the catalytic constant. k_3_ and k_4_ are the binding and the dissociation rate constants for the formation of the EP complex. k_5_ and k_6_ are the microscopic rate constants for the binding (*k_on_*) and the dissociation (*k_off_*) of GAL to the AChE active site.

**Figure 6 ijms-23-05072-f006:**
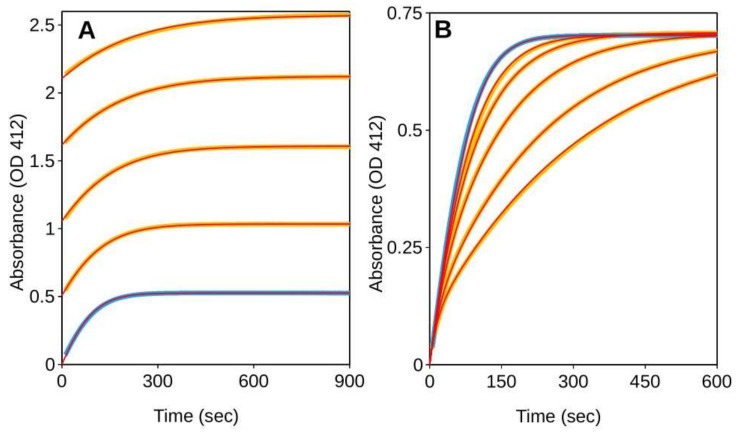
Kinetic analysis of progress curves. Time course of product formation for ATCh and *Tc*AChE in the absence (blue trace) or presence (yellow traces) of the inhibitor. The red lines are from fitting the kinetic parameters to the corresponding experimental data. (**A**) Determination of *k_at_*/*K_m_*, *K_cat_* and of the *K_i_* for product inhibition. Five subsequent reactions were measured in the same cuvette with 200 pM of *Tc*AChE and 35 µM of ATCh. Once the substrate was completely converted to thiocholine-TNB, a new reaction was started by adding further 35 µM of ATCh. Four initial concentrations of thiocholine-TNB were tested (from 35 to ~150 µM). At higher thiocholine-TNB initial concentration the plateau is reached at longer reaction times, which indicates product inhibition. (**B**) Determination of GAL inhibition. *Tc*AChE was 200 pM, ATCh 50 µM and GAL concentrations were 0 (blue trace), 15, 30, 60, 120 and 180 nM (yellow traces).

**Figure 7 ijms-23-05072-f007:**
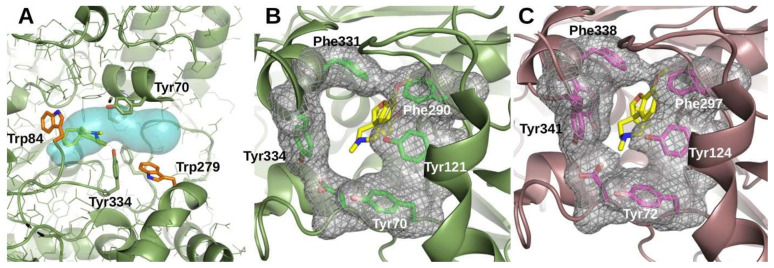
Structure of AChEs active site gorge. (**A**) Depiction of the crystal structure of *Tc*AChE (PDB 1EA5 [46]) showing the internal surface of the active site gorge (cyan, generated using CAVER [45] with a bound GAL molecule (yellow sticks, obtained from superimposition of PDB IDX6 [47]). Four aromatic residues, Tyr70, Phe290, Phe331 and Tyr 334 (green sticks), cause a narrowing of the gorge generally referred to as the bottleneck. Trp 279 and Trp 84 (orange sticks) mark the entrance and the bottom of the gorge, respectively. (**B**,**C**) View of a section of the active site gorge of the apo form of the *Tc*AChE (**B**, PDB 1EA5 [46]) and of the enzyme from human brain (**C**, PDB 4EY4 [48]) seen along the main axis and showing the bottleneck lumen. Residues lining the bottleneck are shown in stick and their van der Waals surface is shown as a gray mesh. Geometric analysis of the gorge performed with CAVER [45] evaluated the opening through the gorge narrowing to have a solvent accessible area of 28 Å^2^ in *Tc*AChE and of 14 Å^2^ in HuAChE. Behind the bottleneck, depicted in yellow stick, is shown the GAL inhibitor bound in the active site at the bottom of the gorge, obtained from superimposition of the AChE-GAL crystallographic complexes PDB ID 1DX6 [47] for *T. californica* and PDB ID 4EY6 [50] for the human enzyme. Created using PyMol [49].

## Data Availability

Not applicable.

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
