# Peer review of "Kinetic Modeling of Time-Dependent Enzyme Inhibition by Pre-Steady-State Analysis of Progress Curves: The Case Study of the Anti-Alzheimer’s Drug Galantamine"

_ijms, 2022, doi:10.3390/ijms23095072_

Round 1

Reviewer 1 Report

The manuscript of Lamba and Pesaresi is well written, the methods are adequate and the conclusions made are based on the results. I have only minor comments:

1) line 48: The ucompetitive should be read uncompetitive

2) lines 52-53: ...termination of the synaptic cholinergic signal transmission...in fact, AChE rather not terminate the neurotransmission as it is localized on the presynaptic membrane (see A. L. Perrier, J. Massoulie and E. Krejci: Neuron 2002 Vol. 33 Issue 2 Pages 275-85) but regulates the transmission

3) lines 76-77: ....tenfold higher Ki...it is 100 times higher (52 vs. 0.52)

Author Response

We gratefully acknowledge Reviewer 1 for the suggested corrections.

It follows a point-by-point reply, with our comments in blue:

1) line 48: The ucompetitive should be read uncompetitive

Done

2) lines 52-53: ...termination of the synaptic cholinergic signal transmission...in fact, AChE rather not terminate the neurotransmission as it is localized on the presynaptic membrane (see A. L. Perrier, J. Massoulie and E. Krejci: Neuron 2002 Vol. 33 Issue 2 Pages 275-85) but regulates the transmission.

The term "termination" has been replaced by "regulation" 

3) lines 76-77: ....tenfold higher Ki...it is 100 times higher (52 vs. 0.52)

The two Ki values in question are of 52 nM and 0.52 µM (i.e. 520 nM). 

Then “tenfold higher” is correct.

Reviewer 2 Report

Dear authors,

The authors presented well designed and conducted research aimed at improving enzyme kinetics methodology. More specifically, authors made an effort to highlight benefits of global fitting of reaction progress curves over simplified initial velocities-based Michaelis-Menten model. Thus, greatest achievement of the work in question is its inspirational character regarding the necessary shift from initial velocities- to progress curves-based analytical methods. However, prior to publication some improvements should be made. The paper in its current form needs to be edited by certified English proofreader. Another major requirement is to reorganize text so that text from lines 500 to 550 in included in discussion section instead of conclusion section. Few minor details are also necessary to address:

  • On few occasions in text Alzheimer’s disease is misspelled as Alzheiner’s
  • Lines 56-57; besides three cholinesterase inhibitors a new drug for the treatment of AD was approved in June 2021 – aducanumab – which targets aggregated Aß so the authors are encouraged to address it in the text
  • Lines 317-318; it should be Ellman’s reagent and not Hellman’s
  • Line 354; shouldn’t it be 35-150 µM of ATCh?

Once the raised questions are adequately addressed, the paper should be accepted for publication.

Best regards     

Author Response

We thank Reviewer 2 for his valuable observations. 

It follows a point-by-point reply with our comments in blue:

1) The paper in its current form needs to be edited by certified English proofreader.

The text has been reviewed by a native English speaker colleague.

2) Another major requirement is to reorganize text so that text from lines 500 to 550 in included in discussion section instead of conclusion section. 

We agree.

The section “Conclusions”  spans lines 493 to 555. Because the few initial and final sentences are merely intended as prelude and as conclusion for the text between lines 500 and 550, we simply moved this entire section under “Discussion”.

3) On few occasions in text Alzheimer’s disease is misspelled as Alzheiner’s

Done.

4) Lines 56-57; besides three cholinesterase inhibitors a new drug for the treatment of AD was approved in June 2021 – aducanumab – which targets aggregated Aß so the authors are encouraged to address it in the text

Aiming to have a concise introduction, we expressly decided to keep the role of acetylcholinesterase in the etiopathogenesis of Alzheimer's disease as brief as possible. For this same reason, we have only barely mentioned rivastigmine, which is a covalent pseudo-irreversible inhibitor of acetylcholinesterase and have completely neglected memantine, which is an antagonist of the (N-Methyl-D-Aspartate)-receptor subtype of glutamate receptor in use for the treatment of Alzheimer's symptoms since year 2000.
Aducanumab is a new therapeutic antibody directed against the ß-amyloid peptide. Studies in mice suggests that it causes a reduction of the amyloid plaques which, in turn, should result in the reduction of neurological and cognitive decline, although on this respect clinical trials have failed to provide sufficient evidences (doi: 10.1002/alz.12213).
Therefore, we feel that deepening, or even just mentioning this subject, would make the already long introduction excessively long without adding significantly to the main topic of the paper. Then, we would rather prefer not to include aducanumab.  

5) Lines 317-318; it should be Ellman’s reagent and not Hellman’s

Done

6) Line 354; shouldn’t it be 35-150 µM of ATCh?

No. In all curves the enzyme experiences an ATCh concentration of 35 µM. This is because a new reaction initiated by adding fresh 35 µM ATCh, started only after that the previously added 35 µM substrate (former reaction) had been completely turn into the thiocholine-TNB product.